# Field-deployable cantilever-enhanced photoacoustic instrument for aerosol light absorption measurement at three wavelengths

Juho Karhu[1,2], Tommi Mikkonen[1], Joel Kuula[1,3], Aki Virkkula[3,4], Erkki Ikonen[2,5], Markku Vainio[1,6], Hilkka Timonen[3], Tuomas Hieta[1,7]

[1]Department of Chemistry, University of Helsinki, Helsinki, FI-00014, Finland
    [2]Metrology Research Institute, Aalto University, Espoo, FI-02150, Finland
    [3]Atmospheric Composition Research, Finnish Meteorological Institute, Helsinki, FI-00101, Finland
    [4]Institute for Atmospheric and Earth System Research, University of Helsinki, Helsinki, FI-00014, Finland
    [5]VTT MIKES, VTT Technical Research Centre Finland, Espoo, FI-02044, Finland
[6]Photonics Laboratory, Physics Unit, Tampere University, Tampere, FI-33014, Finland
    [7]Gasera Ltd., Turku, FI-20520, Finland

*Correspondence to*: Juho Karhu (juho.karhu@helsinki.fi)

**Abstract.** We demonstrate a measurement of aerosol absorption at three wavelengths for particles below 1 µm in diameter, using a highly sensitive photoacoustic spectrometer. The acoustic signal is detected with a cantilever microphone, which
allows sensitive detection without the need to apply acoustic resonance to enhance the signal. The lack of resonator makes the instrument compact and well suited for field measurements. A field instrument employing the method was developed and deployed for black carbon monitoring at an air quality measurement station. The method shows excellent sensitivity for in-situ aerosol absorption measurement, with detection limits of 0.016, 0.025 and 0.041 Mm[-1], for simultaneous measurements at the wavelengths 445, 520 and 638 nm, respectively, using 1 hour averaging time. The black carbon concentration
measured with the new instrument is compared against filter-based photometers operating at the site, showing high correlation.

## 1 Introduction

Black carbon (BC) refers to aerosols composed mainly of elemental carbon, which strongly absorb light throughout the electromagnetic spectrum. It is generated and released into the atmosphere in incomplete combustion processes. Contrary to
most aerosols, which typically have a cooling effect on climate through scattering of sunlight, BC has a warming impact due to high absorption (Bond et al., 2013). As aerosols generally have shorter atmospheric lifetimes, typically days or weeks, compared to many greenhouse gases, controlling BC release into the atmosphere presents a compelling possibility to combat warming of the climate at relatively short time scales (Xu and Ramanathan, 2017). In addition to climate effects, increased BC concentration has a negative health impact, particularly to the respiratory system. It has been suggested that BC
concentration may be an especially good indicator for degraded air quality (Achilleos et al., 2017).

There has recently been increasing interest in improving the quality and coverage of atmospheric BC measurements. There are ongoing efforts by international organizations to establish initiatives to reduce BC emissions, but these are complicated by the lack of standardization of BC measurement methods (Lack et al., 2014; Timonen et al., 2019). Currently the most common methods for real time monitoring of BC are filter-based photometers, where the air sample is continuously drawn through a porous filter and the optical transmission through the filter is monitored over time. As the strongly absorbing BC particles are accumulating on the filter, the decrease in transmission can be related to the BC concentration in the sample flow. These methods are widely used for BC monitoring, but they have major issues arising from the optical properties of the filter material affecting the measurement results (Virkkula et al., 2007). These effects are countered with various correction schemes, but there is significant uncertainty related to the different correction methods and the parameters used in the corrections (Luoma et al., 2021; Savadkoohi et al., 2024).

Methods based on photoacoustic and photothermal effects offer compelling alternatives for BC monitoring, as the absorption measurement is performed directly in the aerosol phase (Moosmüller et al., 1997). These methods are based on the heat released into the surrounding gas, after the BC particles absorb radiation. In photothermal interferometry, the temperature change is recorded as a change in the refractive index (Drinovec et al., 2022), while in photoacoustic spectroscopy, a microphone records the pressure increase resulting from the temperature rise (Petzold and Niessner, 1995). Typically, these methods require bulky instruments to reach high sensitivity: for photothermal interferometry, mechanical stability of the instrument must be especially good for the sensitive interferometric measurement, and photoacoustic instruments typically make use of an acoustic resonance to reach sufficient sensitivity, but acoustic frequencies require relatively long resonators. The bulky size and the use of a resonance, which can be sensitive to environmental conditions, are challenges for the development of sensitive field deployable photoacoustic instruments. However, there have been several field measurement demonstrations, reaching typically single digit $Mm^{-1}$ sensitivity (Arnott et al., 2003; Lack et al., 2012; Linke et al., 2016). A recent article reported a 4-wavelength photoacoustic instrument operated at a remote monitoring station, reaching a detection limit of approximately 0.1 $Mm^{-1}$, with averaging time of half an hour (Schnaiter et al., 2023).

We previously demonstrated the application of cantilever-enhanced photoacoustic spectroscopy (CEPAS) for aerosol measurements (Karhu et al., 2021). CEPAS does not make use of acoustic resonances. Instead, high sensitivity is reached using an optically read cantilever microphone to record the acoustic signal (Kauppinen et al., 2004). The lack of a resonator allows for a compact measurement setup and makes the system particularly suitable for multiwavelength operation: lasers operating at different wavelengths can be multiplexed to different modulation frequencies, freely within the bandwidth of the cantilever microphone. Different wavelength channels can then be measured simultaneously by calculating the Fourier transform of the microphone signal to retrieve its acoustic spectrum. In photoacoustic instruments employing a resonator, the different channels must be recorded sequentially or fitted within the frequency span of the resonance peak, which may be limited and sensitive to environmental conditions.

Our previous CEPAS demonstration operated within laboratory conditions using a single wavelength at 532 nm, reaching a noise level of 0.013 $Mm^{-1}$ with a 20 s measurement time (Karhu et al., 2021). Here we have extended the measurement to three wavelengths over the visible range, packed the measurement setup into a portable instrument suitable for on-site field measurements outside of the laboratory and show its applicability for the first time in field conditions. In the article, we describe the new instrument and evaluate its noise level and stability. We show its capability to monitor absorption at three wavelengths with high sensitivity when operated at an air quality measurement station in Helsinki, Finland. Furthermore, we demonstrate that the BC concentration measured with the instruments compares well against two reference BC instruments operating at the station. We also show that the new instrument can be applied for sensitive detection of $NO_2$, simultaneously with the aerosol light absorption measurement.

## 2 Experimental

The cantilever-enhanced photoacoustic detector for single wavelength operation has been described previously in (Karhu et al., 2021). The new field-deployable setup with three wavelength channels and revised sampling system is presented in Figure 1. The photoacoustic cell is based on the cell from a PA201 gas analyser (Gasera). It is 10 cm long and 4 mm in diameter. The cell is made of aluminium and coated with nickel. The cell is originally intended for trace gas measurements so the original valves, which were tested to have relatively high particle losses due to small diameter lines and sharp turns, are replaced with solenoid valves designed for high flow rates (LHD, The Lee Company). The end windows are fused silica planar windows with a coating that is antireflective over the visible wavelengths, as well as hydrophobic and oleophobic (Hydrophobic Windows, Edmund optics). The light source is an RGB laser module (µRGB module, OptLasers), where the output consists of three superimposed beams from multimode diode lasers emitting at 638, 520 and 445 nm. Going forward, we will refer to the signals measured with the different lasers as the red (638 nm), green (520 nm) and blue (445 nm) channels. The maximum total optical power from the laser module is 6 W, but the total power used in the experiments was kept substantially lower for better stability and easier thermal management. Each laser diode is connected to a common baseplate, which is cooled with a thermoelectric cooler. The beam is directed through an iris and focused in the middle of the photoacoustic cell. The photoacoustic signal is recorded with an optically read cantilever microphone located in the middle of the cell. Changes in the optical powers are monitored after the cell with a silicon photodiode. The beam after the cell is sampled with a fused silica plate and then further attenuated with a neutral density filter before reaching the photodiode. The photodiode was only used to track changes in the optical power, and the initial power at each wavelength was measured with a thermal power meter (PM160T, Thorlabs) during calibration. The initial powers, as measured after the photoacoustic cell, were 129, 211 and 301 mW for red, green and blue channel, respectively.

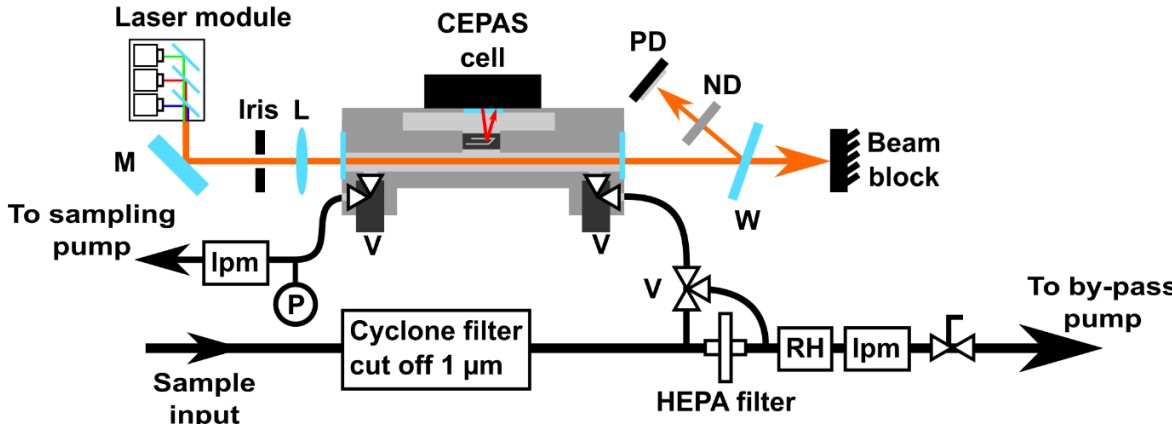

**Figure 1: Schematic picture of the 3-wavelength CEPAS instrument. The superimposed beams from the laser module are directed through the CEPAS cell with a mirror (M) and a lens (L). After the cell, the optical power is monitored by sampling the beam with a wedged optical window (W) and attenuating the sampled beam with a neutral density filter (ND) before it is incident on a silicon photodiode (PD). There is a constant by-pass flow through the instrument, from which the sample is drawn to the CEPAS cell either before (sample measurement) or after (background measurement) a HEPA filter. The gas flow is controlled with solenoid valves (V). Flow sensors (lpm) monitor the by-pass and sampling flow rates and the humidity of the by-pass flow is also measured (RH). The sample pressure (P) is recorded from the sampling line after the CEPAS cell.**

Power of each laser diode can be modulated individually with an analogue signal fed to the laser driver. We used square waves with frequencies 60, 70 and 80 Hz for the modulation of red, green and blue channels, respectively. The duty cycle of the modulation was 50 % for all the channels. The bandwidth of the cantilever microphone used here is up to approximately 700 Hz. The modulation frequencies were chosen from a range where the microphone noise spectrum was free of any obvious noise peaks arising from external noise sources. The channels were measured simultaneously by taking a 1.5 s block of the microphone signal and calculating its Fourier transform. The height of the peak in the spectrum at each modulation frequency was taken as the signal for that channel. The optical power of the individual channels was monitored the same way but using the photodiode signal.

The CEPAS device was calibrated using known absorption cross section of $NO_2$ at each wavelength. The calibration procedure is described in more detail in Appendix A. It should be noted that, while $NO_2$ calibration is widely used in photoacoustic black carbon measurements, it has several limitations. For example, the low absorption cross section at longer wavelengths affects calibration of the red channel specifically, and uncertainty in laser wavelengths imposes uncertainty in converting the $NO_2$ absorption cross section to absorption value for each channel. Calibration based on dye particles (Foster et al., 2019) or a simultaneous extinction measurement could help improve accuracy of the results (Arnott et al., 2000).

The CEPAS cell must be closed during the measurement, so the sampling is done in sample-and-hold configuration. Sample is periodically drawn into the CEPAS cell from a by-pass flow of flow rate 2 l/min, as monitored with a flow meter (D6F, Omron). During the sample exchange, the flow through the CEPAS cell is approximately 0.5 l/min, which was also monitored with a second flow meter (D6F, Omron). Each sample is measured for 15 s. The total measurement cycle, including the sample exchange, takes approximately 20 s. The humidity of the by-pass flow is measured with a humidity sensor (ChipCap 2, Amphenol Telaire). The final measurement pressure inside the CEPAS cell is close to atmospheric pressure and it is recorded with a pressure sensor, just before closing the cell valves. The CEPAS cell temperature is slightly elevated with resistive heating to 45 °C. This is mostly to decrease relative humidity inside the cell to ensure no water condensation can take place, although in these experiments all the samples have low humidity to begin with. The microphone signal and pressure reading are digitized by electronics included in the original PA201 gas analyser and read to a PC via USB. All other sensors are digitized with a data acquisition card (USB-6351, NI), which is also used to generate control signals for the laser and the gas exchange system. Control and data processing on the PC is done via a custom LabView program.

The input of the by-pass flow has a cyclone filter with a cut-off diameter of 1 µm for particles. After 5 measurement cycles, two particle-free samples are taken from the same by-pass line, but after a HEPA filter, to record any changes in the background level. It should be noted that the background level includes any gaseous absorption in addition to absorption signal arising from windows or light hitting the walls of the CEPAS cell. The background signal is linearly interpolated between the background measurements and subtracted from the normal signal measurements.

The device and accompanying electronics are packed into a field prototype consisting of two rack enclosures. The box for the electronics is one rack unit (1U) high and the other one containing the optical setup is 2U high. The stability of the device was tested with an overnight measurement of laboratory air, which was filtered through a HEPA filter. The instrument was moved to the air quality monitoring station SMEAR III in Helsinki (Järvi et al., 2009). The station is classified as urban background station and typical BC concentrations are relatively low, with the average BC concentration over the measurement period of approximately 200 ng/m³. The station is located at a university campus and surrounded by sparse buildings and low vegetation, with closest major road at a distance of approximately 100 m. The sample flow drawn from the outdoor air is dried with Nafion tubing. The BC concentration measured with CEPAS are compared against two filter-based photometers operating at the station: an aethalometer AE33 (Aerosol Magee Scientific) and a multi angle absorption photometer MAAP (Thermo Scientific).

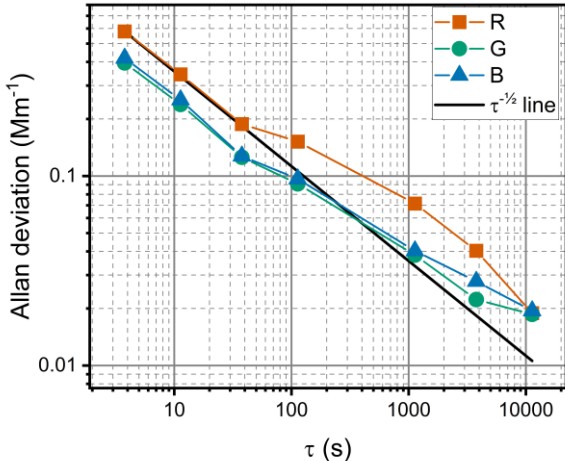

**Figure 2: Allan deviation of CEPAS absorption measurement for each channel. The figure also shows a line corresponding to white noise averaging (black line), where the Allan deviation decreases as a function of the square root of the averaging time. The modulated optical average powers were 129, 211 and 301 mW for red, green and blue channel, respectively.**

## 3 Results

### 3.1 Laboratory evaluation of noise and stability

The stability and sensitivity of the instrument were tested with an overnight measurement of laboratory air filtered through an external HEPA filter as a blank sample. Figure 2 shows the Allan deviation (Werle et al., 1993) calculated from the long measurement of filtered air for each wavelength. The first two hours after the instrument was turned on were not included in the stability analysis due to the instrument warming up. The total measurement time after the warmup was 14 hours. When the data was averaged over each sample exchange cycle of 20 s, the standard deviations over the whole data set were 0.28, 0.18 and 0.19 $Mm^{-1}$ for the red, green and blue channels, respectively. The time step between measurement cycles is 26 s seconds on average, which is slightly longer than the time it takes to complete one cycle, because the background measurements are not included in the data. Each channel also showed a small negative offset (-0.12, -0.13 and -0.19 $Mm^{-1}$ for the red, green and blue channels, respectively), which is most likely related to a small pressure difference between the sample and background measurement cycles, as they are sampled from different points in the by-pass flow and from different sides of the HEPA filter. Overall, the Allan deviation shows good stability for the blank measurement, with all channels close to white noise averaging behavior. The noise level is higher compared to our previous laboratory demonstration (Karhu et al., 2021), by approximately one order of magnitude. This is largely due to lower laser power in the portable instrument, which accounts for approximately a factor of 5, but isolation of mechanical noise from the microphone is also slightly worse compared to the laboratory setup. The short term noise is 0.58, 0.40 and 0.42 $Mm^{-1}$ for red, green and blue channels, respectively. Each point in the raw data corresponds to the signals calculated from a single 1.5 s time block

used in the Fourier transforms, but the average time step for the raw data used to calculate the Allan deviation is 3.8 s. This includes all the dead time originating from sample exchange and background measurements, as well as signal processing

between each measurement step." With 1 hour averaging time, the Allan deviation is below 0.05 Mm$^{-1}$ for all channels.

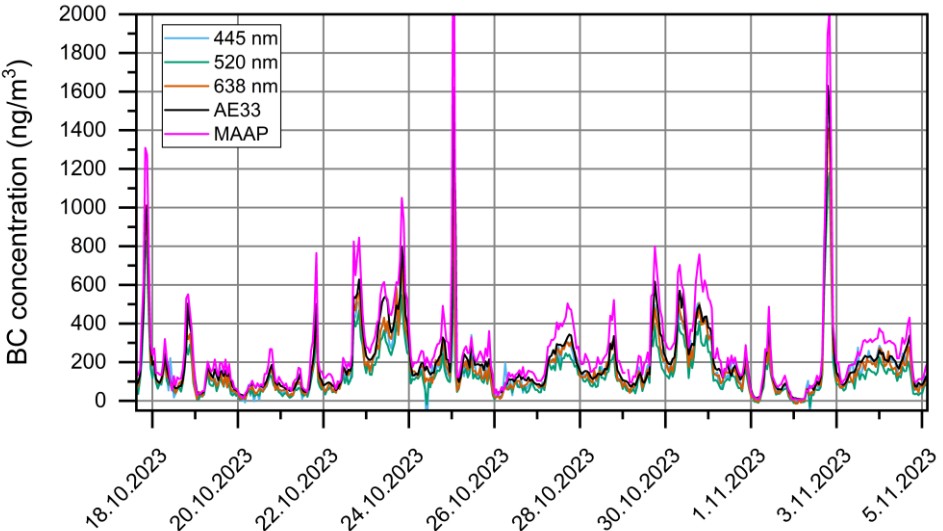

**Figure 3: Time trace of BC measurements at SMEAR III station for each wavelength channel of the CEPAS instrument. Figure also shows BC concentration from two filter-based photometers (AE33 and MAAP) for comparison. The concentration for AE33 is taken from the BC6 channel, measured at 880 nm. All traces are 1-hour averages.**

**3.2 Field measurement of ambient BC**

Figure 3 shows the CEPAS measurement from SMEAR III station, presented as BC concentration. The absorption values were converted to mass concentration by dividing with mass absorption cross section, using values 6.6, 8.1 and 9.5 m$^2$/g for the red, green and blue channels, respectively. The value for the red channel is the same that is used with the MAAP at the wavelength 637 nm and the other two are scaled from it assuming an absorption Ångström exponent of 1. The CEPAS

measurement was also corrected for particle losses. The particle transmission from the input of the field prototype to the output of the CEPAS cell was measured to be approximately a constant 0.7 for particles up to 500 nm (details are described in the Appendix B), and the BC concentration measured with CEPAS at the air quality station was corrected by dividing the results with this transmission. The concentrations are shown as 1-hour averages. The results show excellent agreement with concentrations measured with two filter-based photometers (MAAP, Thermo-Scientific and AE33, Aerosol Magee

Scientific) that were operating on the station during the measurement period. Overall, the reference instruments show slightly larger concentrations compared to CEPAS, which is a typical result when comparing filter-based photometers and photoacoustic instruments (Arnott et al., 2003; Davies et al., 2019; Schnaiter et al., 2023; Zhao et al., 2020). The deviation is usually accounted for uncertainty in the filter corrections, such as the multiple scattering correction. The two reference instruments can be seen to deviate from each other as well, but it should be noted that the values used here are direct

readings from the instruments, without using any site-specific corrections, and used here more to illustrate the excellent

correlation rather than accuracy. At several occasions, the blue and green CEPAS channels deviate from the red channel and the other instruments for short periods, appearing as sharp peaks in the data. These are mostly related to fast changes in the $NO_2$ concentration, which the background subtraction was not fast enough to compensate for, since the background is only updated approximately every two minutes. We did not observe major variations in the ratio between the three channels

throughout the measurement, suggesting no significant changes in the absorption Ångström exponent over the measurement period (see appendix C for details).

Figure 4 shows the linear regressions of the 1-hour averaged BC concentration measured with CEPAS against the concentration measured with AE33 and MAAP. For this comparison, we have used the AE33 channels with wavelengths

closest to each CEPAS channels. The plot for red the channel shows one clear outlier below the linear fit, with concentration of approximately 1000 ng/m$^3$ according to AE33. This corresponded to the decreasing edge of the sharp peak on 25.10., which is apparent in Figure 3. The two other channels show a few more outliers related to the $NO_2$ interference as described above. All the outliers are included in the fits and the statistics shown in Figure 4. The Pearson's correlation coefficient between the concentration measured with CEPAS and AE33 was 0.968, 0.964 and 0.955 for the CEPAS red, green and blue

channels, respectively. The correlation coefficient between the concentration measured with MAAP and with CEPAS red channel was 0.975, showing slightly better correlation compared to the AE33 comparison. For comparison, the correlation coefficient between AE33 and MAAP data shown in Figure 3 was 0.974. The root mean square errors of the fits against the AE33 data are 43.8, 37.0 and 51.0 ng/m$^3$, for red, green and blue channels, respectively. The root mean square error of the fit against the MAAP measurement is 39.2 ng/m$^3$. The difference in the slopes of the fits from the different channels suggest

that there is some disagreement in the wavelength dependence of absorption between the instruments, although at least part of this could be attributed to uncertainty in the $NO_2$ calibration of CEPAS.

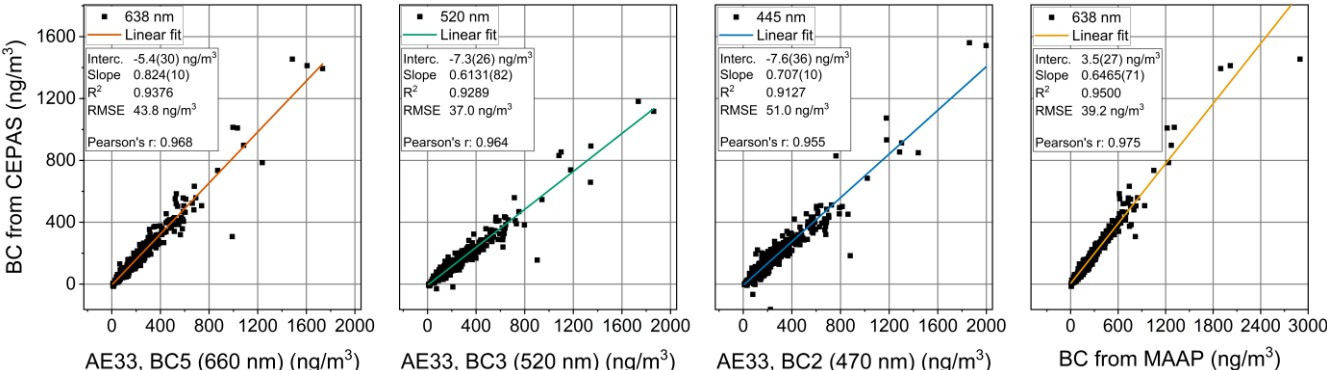

**Figure 4: BC concentration measured with the three CEPAS channels, as a function of the concentration measured with AE33 and MAAP filter-based photometers. The text box shows the fit parameter with the statistical uncertainty (1$\sigma$) of the least significant**

**digits in parenthesis, as well as the coefficient of determination ($R^2$) and root mean square error (RMSE) of the fit, and Pearson's correlation coefficient between the datasets (Pearson's r).**

Figure 5 shows a closeup to a small part of the time trace to illustrate the agreement at both low and high concentrations. A shorter averaging period of 5 min is used here. The period from 2.11. 0:00 to 6:00 shows a BC concentration close to zero and could be used as a more realistic estimate for the detection limit. The average concentration over this period given by AE33 is 11 ng/m$^3$ and the concentration given by CEPAS is -6.3, -5.8 and 0.0 ng/m$^3$ according to the red, green and blue channels, respectively. The standard deviations of absorption measured with CEPAS over this period are 0.14, 0.11 and 0.11 Mm$^{-1}$ for the red, green and blue channel, respectively, using 5-minute averaging. With the longer 1-hour averaging shown in Figure 3, the standard deviations over the same time frame are 0.041, 0.025 and 0.016 Mm$^{-1}$, which match well with the Allan deviations from Figure 2, although it should be noted that with 1-hour averaging, the interval only contains 6 measurement points for each channel. Some of the occasional sharp outliers in Figure 5 are due to the NO$_2$ interference as described above, particularly when green and blue channels can be seen to deviate from the red channel, but some of the peaks are likely erroneous readings from the microphone. These outliers were included in all the statistics described above.

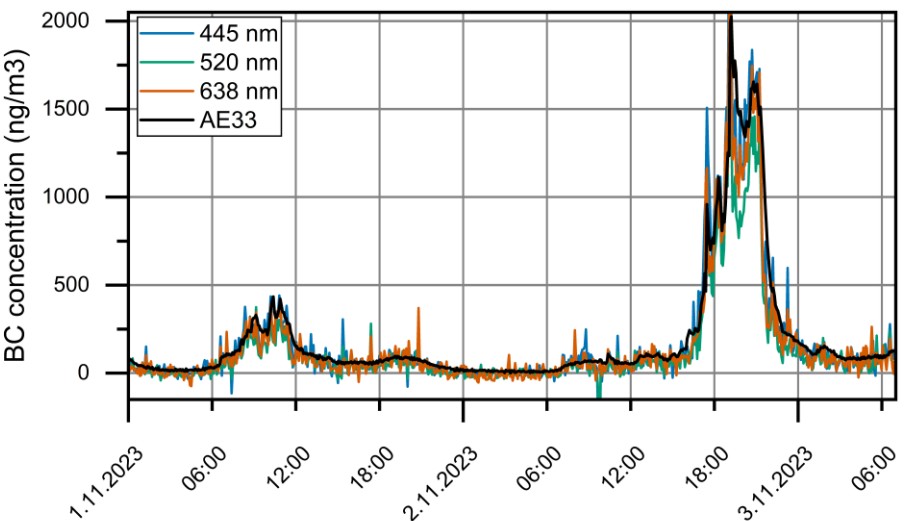

**Figure 5: Part of the time trace of the SMEAR III measurement with the three CEPAS channels and with AE33 for comparison. All traces are 5-minute averages.**

## 3.3 Simultaneous NO$_2$ concentration estimate

The background signal, which is measured after the sample flow passes through a particle filter, consists of an instrumental background, which is influenced by factors such as residual absorption by the cell windows, and a gaseous absorption background due to absorption signal from atmospheric gases. The main contributor to the gaseous absorption component at the wavelengths used for the measurements is NO$_2$. The instrumental background can change over time due to effects such as drifting of the laser alignment or accumulating of absorbing particles on the cell windows. The average drift of the

instrumental background per day over the measurement period was measured to be approximately 6.5, 10 and 7.7 Mm$^{-1}$, for red, green and blue channels, respectively, and it was observed to be mostly correlated between the channels. Since the NO$_2$ absorption affects the background signals of the measurement, we can also estimate changes in the NO$_2$ concentration during the BC measurement. At typical atmospheric concentrations, NO$_2$ absorption at the red wavelength is almost negligible compared to the other two wavelengths (Vandaele et al., 1998), so relative changes in the background signal of the three wavelengths can be used to estimate changes in the NO$_2$ absorption. For example, for the blue channel we can write:

$$S_{NO2}^{B} = S^{B} - k_{R}^{B} \times S^{R}. \tag{1}$$

The background signals for the red and blue channels ($S^{R}$ and $S^{B}$, respectively) are both influenced by effects such as particles accumulating on the windows, but only $S^{B}$ has significant contribution from NO$_2$ absorption. If we approximate that the changes other than the NO$_2$ absorption in both backgrounds are proportional to each other, we can scale the background signal of the red channel with a factor $k_{R}^{B}$ and subtract it from the other two. This effectively attempts to account for changes in the instrumental background of the blue channel, according to how the red channel background has changed. The remaining background after the red channel subtraction ($S_{NO2}^{B}$) is assumed to be due to NO$_2$. This can be converted to NO$_2$ concentration according to the NO$_2$ calibration. Similar equation applies for the green channel, but with its own factor $k_{R}^{G}$.

The assumption that the background signals, excluding the NO$_2$ contribution, are proportional to each other appears to hold well for several hours, but on longer time scales, they start to drift apart from each other. For proper NO$_2$ measurements, a secondary background measurement with NO$_2$ free sample should be performed approximately daily, to adjust the scaling of the red channel subtraction. This could be achieved, for example, with bottled air or ambient air with a suitable adsorbent to remove NO$_2$. However, here we adjusted the scaling factor afterwards, by occasionally matching the concentration reading against a reference NO$_X$ instrument operating at the station, choosing points when NO$_2$ concentration was low and stable. The time between these points was typically one or two days. The scaling factor $k_{R}^{B}$ was adjusted at these anchor points, so that the concentration given by CEPAS and the reference instrument were equal, and $k_{R}^{B}$ was then linearly interpolated between these points. Figure 6 illustrates that the changes in the NO$_2$ concentration outside of these anchor points are reproduced well with this method. In the figure, the NO$_2$ concentration measured with CEPAS is the average of the calculations from the blue and green channels.

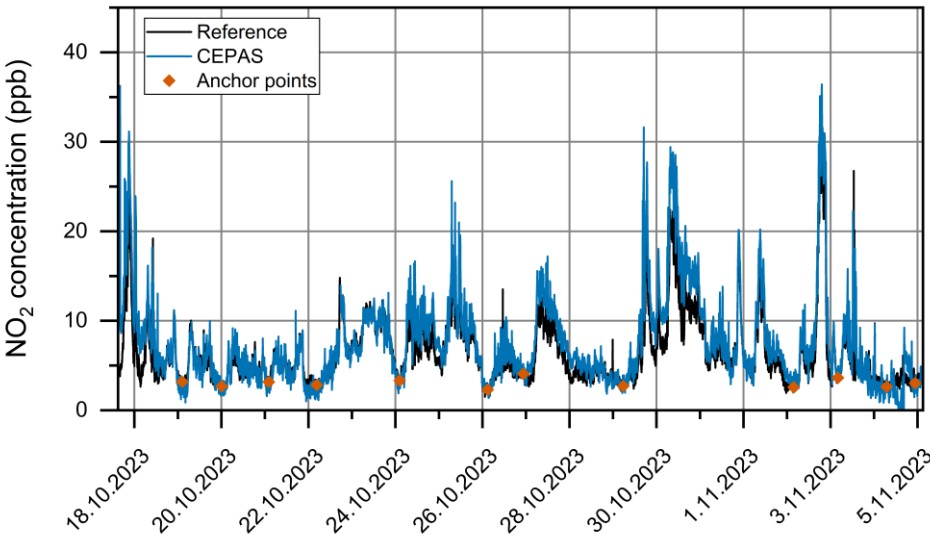

**Figure 6: NO₂ concentration calculated from the CEPAS background signals. NO₂ concentration measured with a chemiluminescence instrument is shown for reference. To account for background drifting, the NO₂ concentration measured with the two instruments was matched at the marked anchor points. Averaging time for both data is 5 min.**

**4 Conclusions**

Our results demonstrate that CEPAS is well suited for multiwavelength measurement of aerosol absorption and performs well as a field instrument. Due to the high sensitivity of CEPAS, we could reach detection limits of 0.016, 0.025 and 0.041 Mm$^{-1}$ for absorption at the wavelengths 445, 520 and 638 nm at 1 hour averaging time, without using an acoustic resonator. The high sensitivity also allowed for simultaneous measurement of NO₂ trace concentration at ppb-level. The performance of CEPAS shows particular promise towards applications where measurements of low concentrations are desired, such as clean environments and size resolved measurements of BC, as demonstrated in a parallel study (Kuula et al., 2024).

The detection limits demonstrated here are somewhat degraded compared to our earlier laboratory demonstration (Karhu et al., 2021), mainly due to lower optical power used per laser channel. The total power of approximately 0.65 W is at a level similar to the previous demonstration, but it is now divided between the three channels. The laser power per channel was kept at a lower level mostly for easier thermal management of the enclosed portable system. The new laser is also a compact and inexpensive multimode laser module with a significantly worse beam quality compared to the single mode laser used in the previous laboratory demonstration. The light from the beam edges coming in contact with the cell walls increases the background level, which means that the laser instability will start limiting the measurement noise at lower power. There is potential for further improvement in the detection limit by increasing the laser power, but it may require improving the thermal management of the instrument to ensure stable laser operation or changing to a more costly single mode laser

module to reduce the background level. Improving the mechanical noise isolation can also be expected to improve the sensitivity by up to a factor of 2.

Although the repeated background measurements compensate for the $NO_2$ interference adequately, a fast change in the $NO_2$ concentration can still occasionally produce erroneous signals between two background measurements. A potential further complication is that for several BC sources, traffic for example, $NO_2$ concentration is likely to change simultaneously with BC, although this did not appear to be a noticeable problem in our measurement set. A straightforward improvement would be to simply measure the particle-free background more frequently. The limiting case would be to alternate between consecutive signal and background measurements, which would effectively double the response time of the instrument. Alternatively, $NO_2$ could be removed from the sample flow using a suitable absorbent (Arnott et al., 2003). Another solution could be to use a differential measurement, where the particle sample is introduced on one side of the cantilever and a filtered, particle free sample on the other, while both sides are illuminated with the lasers. If the absorption due to any gas species is equal on both sides, it cancels out at the microphone. Such methods have been demonstrated in gas sensing with a cantilever microphone, although in those reports, the microphone is essentially used as a wavelength specific power detector (Uotila, 2007). The issue with a direct gas measurement is that gas can flow pass the micrometer-scale gap around the cantilever, so that the gas concentration will slowly equalize between the two sides of the cantilever. However, the method might be suitable for measurement of particles, which are less likely to pass through the gap during a measurement, although the differential measurement would come at a significant cost of system complexity.

We have also shown that the setup can be packed into a portable instrument, and it performs well in field measurements at an air quality station. The detection limit estimated from the field measurement dataset matches well with the results from the laboratory noise test. The instrument size is already well portable, at 3 rack units, and the method is well suited for further miniaturization. For example, we are using a rather bulky data acquisition card for data recording and computer control, which could be replaced with a more compact alternative. A significant advantage of the method is that, since no acoustic resonances are needed to amplify the signal, the size of the photoacoustic cell can be kept compact. The size and portability of available instruments are important factors in expanding the coverage of BC measurements in the future.

**Appendix A: $NO_2$ calibration**

The CEPAS response must be calibrated using a sample with known absorption in the photoacoustic cell. Here, we have used $NO_2$ gas for the calibration, which is a common option for measurements at visible wavelengths. However, it should be noted that there are several drawbacks with this calibration method. The $NO_2$ absorption cross section is very wavelength dependent, and there are some disagreements between spectra reported in the literature. We have used a spectrum from (Vandaele et al., 1998) accessed through the MPI-Mainz UV/VIS Spectral Atlas (Keller-Rudek et al., 2013). This spectrum

is, for example, basis for the JPL recommended reference spectrum (Burkholder et al., 2019). Another issue is the relative signal levels at 445 nm and 638 nm: the absorption cross section at 638 nm is significantly smaller, making it difficult to calibrate all the channels at the same time. Calibrating one laser channel at the time, on the other hand, results in the temperature of the laser module drifting significantly from the typical operational temperature, leading also in drifting of the laser power and wavelength.

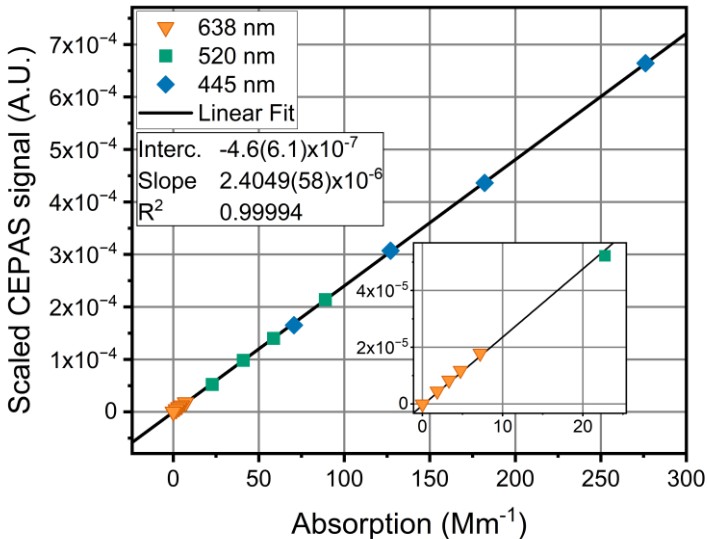

**Figure A1: CEPAS signal, scaled by optical power and frequency response at each channel, as a function of the NO₂ absorption. The scaling allows applying all the calibration data measured at the different wavelengths to the same calibration fit. The NO₂ absorption is calculated for the different wavelengths from literature NO₂ absorption cross section, at each NO₂ concentration steps. Values scaled from the red, green and blue channels are shown with different coloured symbols.**

Here, we perform the calibration for each channel at the same time, but instead of calculating individual fits for each channel, we scale the CEPAS response into one single fit. Similar method was used for example in (Schnaiter et al., 2023). The photoacoustic signal at each wavelength channel is a function of the optical power at that wavelength, as well as the modulation frequency used for the channel. The optical power at each wavelength was measured after the photoacoustic cell with a thermal power meter (PM160T, Thorlabs), by switching each laser on one at a time. The frequency response was measured by turning only the blue laser on and recording the photoacoustic background signal resulting from the residual window absorption at each modulation frequency. The CEPAS signal from each channel and for each calibration step was normalized by the optical power and the microphone frequency response. Figure A1 shows the normalized CEPAS signal plotted as a function of the absorption calculated from the $NO_2$ concentration and absorption cross section. Four known concentrations of $NO_2$ were diluted from a gas cylinder with 1.17 ppm of $NO_2$ in $N_2$ balance. The diluting gas was compressed air. The concentration levels were converted to absorption at each wavelength using the absorption cross section from the literature (Vandaele et al., 1998). The laser wavelengths were measured with an optical spectrum analyser

(AQ6315E, Ando). Figure A1 shows that the scaled CEPAS data follows well a linear fit. To get a conversion factor for each channel, the slope from the linear fit was again scaled with the optical powers and frequency response. These factors were then used to convert the CEPAS signals to absorption in the BC measurements.

To estimate the uncertainty of the absorption measured with CEPAS, we consider contribution from the $NO_2$ concentration and absorption cross section used for the calibration and from the power measurement used to scale the CEPAS signals from the different channels. The uncertainty of the power meter is given as 5%. The $NO_2$ concentration was varied with a mass flow controller (FC-785C, Aera) and we estimate the uncertainty of the dilution system to be 5% based on the accuracy of the mass flow controller. Since the $NO_2$ absorption cross section here is based on literature values, we estimate its
uncertainty by comparing several spectra (Davidson, et al., 1988; Vandaele et al., 1998; Vandaele et al., 2002; Voigt, et al., 2002) available from MPI-Mainz UV/VIS Spectral Atlas (Keller-Rudek, et al., 2013). The standard deviation of the cross section at the laser wavelengths from the four different spectra gives relative uncertainty of 9.8%, 1.6% and 4.0% for the red, green and blue channels, respectively. The $NO_2$ absorption is further influenced by any wavelength changes of the three lasers. Based on wavelength measurements performed in laboratory conditions on different days, we estimate that the
wavelength of the lasers may vary within approximately a range of 1 nm. Allowing each laser wavelength to vary by 0.5 nm to either direction imparts a relative uncertainty of 2.9%, 3.5% and 4.8% for the red, green and blue channel, respectively. The red channel has the largest uncertainty for the absorption cross section, dominated by the differences between different sources. If we use 10% uncertainty for the absorption cross section, combining the other sources addressed above, we get a total uncertainty of 12% for the absorption measured with CEPAS.

**Appendix B: Particle loss measurement**

A scanning mobility particle sizer system was used to estimate the particle losses of the CEPAS measurement. Particles were generated with an atomizer from an $NH_4NO_3$ water solution and size selected with a differential mobility analyser (DMA). From the DMA output, the flow was directed to the CEPAS instrument, inside of which the flow was divided into the 2 l/min by-pass flow and the sample flow going to the photoacoustic cell. In the typical measurements, the sample flow through the
370 cell during the sample exchange was approximately 0.5 l/min, but in the loss measurement, we used a condensation particle counter (CPC) to draw a 0.3 l/min sample flow through the cell. The particle concentration measured by the CPC was then compared to a separate measurement performed directly from the DMA output to estimate the particle transmission through the photoacoustic cell. The DMA sheath flow was 3.5 l/min. DMA was tuned to five different particle sizes, ranging from 100 nm to 700 nm. No inversion was calculated for this estimate. Figure B1 shows the measured transmission as a function
of particle diameter. The figure shows that the losses are approximately constant, with a transmission of approximately 0.7, until 400 nm, after which the losses start to increase for the larger particles. Since we did not have a measured particle size distribution specific for BC in the concentration measurement of the main manuscript, we have used this transmission of 0.7

to correct the measured BC concentrations, as BC particles are typically smaller than 500 nm in size. The result can be thought of as the higher limit for the particle losses, because the particle concentration is measured at the output of the CEPAS cell, so it includes possible losses at the outcoupling from the cell, which would not affect the measured CEPAS signal.

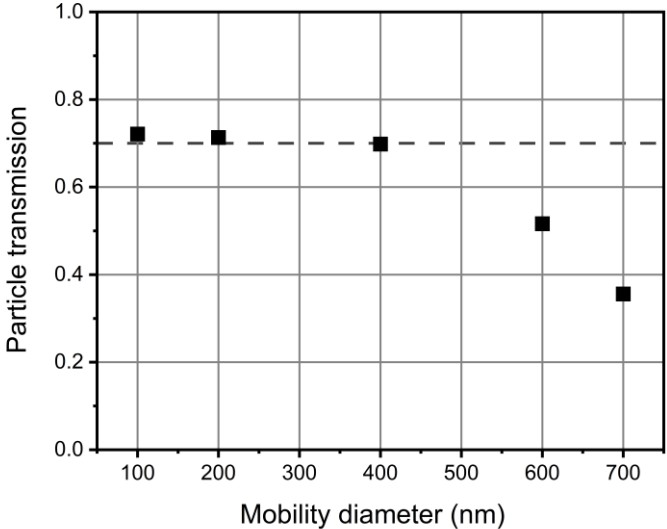

**Figure B1: Particle transmission measured with SMPS from input of the CEPAS instrument to the output from the CEPAS cell. The figure shows that the transmission is nearly constant at approximately 0.7 (dashed line) below diameter of 400 nm, but starts to decrease for larger particles.**

## Appendix C: Absorption Ångström exponent

The absorption Ångström exponent (AAE) calculated from the CEPAS red and blue channels, as well as from AE33 channels with nearest matching wavelengths (BC5 at 660 nm and BC2 at 470 nm) are shown as a time trace in figure C1. The data points are 1-hour averages. The figure C1 also shows the correlation between the AAE measured by the two methods. The concentration from AE33 were converted to absorption by multiplying with the mass absorption coefficient from the instrument manual (10.35 $m^2$/g for BC5 and 14.54 $m^2$/g for BC2). When the BC concentration goes to near zero, the AAE from CEPAS becomes exceedingly noisy, so we have left out the CEPAS data for points where the BC concentration is below 100 ng/$m^3$, as measured by the CEPAS red channel. There are no significant changes observed in the AAE over the measurement period and so no clear correlation is observed either. The average AAE from CEPAS is 0.96, with a standard deviation of 0.46, and the mean from AE33 is 1.46, with a standard deviation of 0.12.

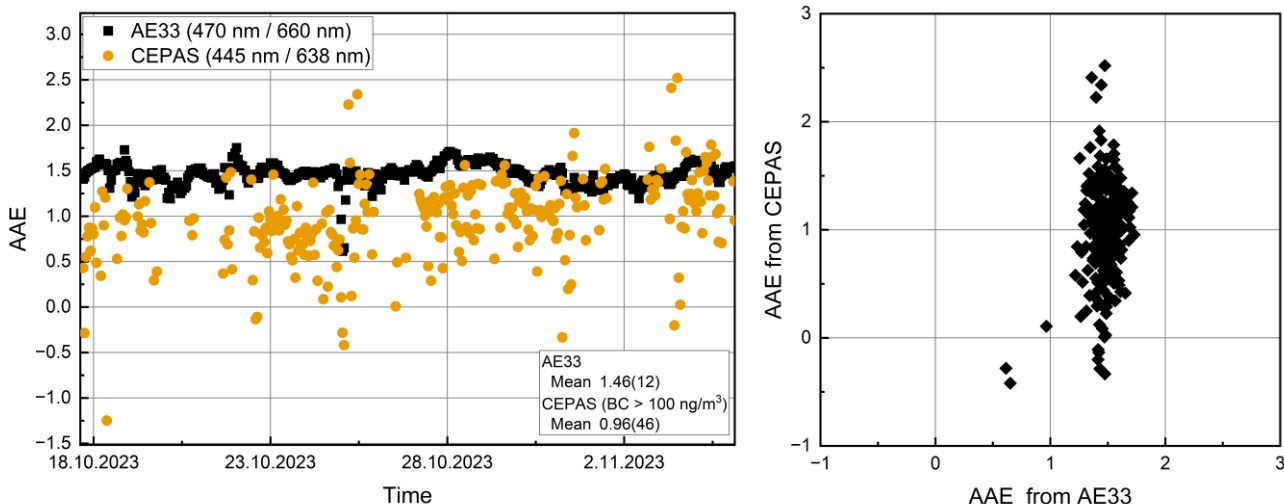

**Figure C1: The AAE measured by AE33 and CEPAS over time (left) and the correlation between the AAE measured by the two methods (right). Each data point is a 1-hour average. For clarity, the farthest outlier has been left outside of the axis limits on the correlation plot, but is shown in the time trace.**

### Data availability

The BC concentration from MAAP and the $NO_2$ reference instrument data are available through the SmartSMEAR (Junninen et al., 2009) web interface available currently at https://smear.avaa.csc.fi/. Other underlying data is available at https://doi.org/10.5281/zenodo.14496882 (Karhu et al., 2024).

### Author contribution

JKa, TM, MV and TH developed and built the photoacoustic instrument; All co-authors planned the measurements; JKa, TM and JKu performed the measurements; JKa and TM analysed the data; JKa prepared the manuscript draft; All co-authors reviewed and edited the manuscript.

### Competing interests

Some authors are members of the editorial board of journal Aerosol Research.

**Acknowledgements**

This work was supported by Research Council of Finland (decisions 341271, 342579, 349544, 361835), Jane and Aatos Erkko Foundation (Project: Compact and precise sensor for global BC monitoring), Business Finland (decision 6868/31/2022), and Research Council of Finland Flagship Programme, Photonics Research and Innovation PREIN (decision 346529).

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
