# Peer review of "Field-deployable cantilever-enhanced photoacoustic instrument for aerosol light absorption measurement at three wavelengths"

_Aerosol Research, 2024_

## Author Comment (AC1)

We thank the reviewers for their thoughtful comments, which we have used to improve the manuscript. We have addressed the specific concerns of the reviewers in the remarks below. The reviewer comments are in bold font for clarity. Any changes made to the manuscript are in italics and the location of the revisions have been marked with the liner number in the revised manuscript.

**Anonymous Reviewer 1 comments:**

**Abstract:**
**- please state that instrument measures absorption of aerosol smaller than 1 μm.**
Abstract was revised accordingly.

The original sentence "*We demonstrate a measurement of aerosol absorption at three wavelengths using a highly sensitive photoacoustic spectrometer.*"

was revised to (line 13) "*We demonstrate a measurement of aerosol absorption at three wavelengths for particles below 1 μm in diameter, using a highly sensitive photoacoustic spectrometer.*"

**- in my opinion the term "field instrument" would be more accurate than "portable instrument".**
Revised as suggested.

**Results:**
**- please include a correlation plot of the absorption coefficient measured using MAAP and CEPAS at 638 nm.**
Correlation plot against MAAP was added as a new subplot in figure 4. A short discussion was added to the second paragraph of the section 3.2 regarding the correlation.

The text in the original manuscript: "*The Pearson's correlation coefficient between the concentration measured with CEPAS and AE33 was 0.968, 0.964 and 0.955 for the CEPAS red, green and blue channels, respectively. For comparison, the correlation coefficient between AE33 and MAAP data shown in* **Error! Reference source not found.** *was 0.974. The root mean square errors of the fits are 43.8, 37.0 and 51.0 ng/m³, for red, green and blue channels, respectively.*"

was revised into (line 208): "*The Pearson's correlation coefficient between the concentration measured with CEPAS and AE33 was 0.968, 0.964 and 0.955 for the CEPAS red, green and blue channels, respectively. The correlation coefficient between the concentration measured with MAAP and with CEPAS red channel was 0.975, showing slightly better correlation compared to the AE33 comparison. For comparison, the correlation coefficient between AE33 and MAAP data shown in* **Error! Reference source not found.** *was 0.974. The root mean square errors of the fits against the AE33 data are 43.8, 37.0 and 51.0 ng/m³, for red, green and blue channels, respectively. The root mean square error of the fit against the MAAP measurement is 39.2 ng/m³.*"

The caption of figure 4 was also revised accordingly.
The sentence "*as a function of the concentration measured with AE33 filter-based photometer.*"
was revised to: "*as a function of the concentration measured with AE33 and MAAP filter-based photometers.*"

The reference to figure 4 in the second paragraph of section 3.2 was also revised from:
"**Error! Reference source not found.** *shows the linear regressions of the 1-hour averaged BC concentration measured with CEPAS and AE33.*"

into (line 203):

"***Error! Reference source not found.*** *shows the linear regressions of the 1-hour averaged BC concentration measured with CEPAS against the concentration measured with AE33 and MAAP.*"

**- please include a correlation plot of absorption Angstrom exponent measured with AE33 and CEPAS.**

A figure showing a time trace of the AAE measured with CEPAS and AE33 and their correlation was added as an appendix C, together with a short discussion. However, we did not observe significant changes in the AAE, so the correlation remains highly uncertain. A reference to the new appendix was added to the end of the first paragraph is section 3.2 (line 201).

The discussion in the appendix reads (line 386):

"*The absorption Ångström exponent (AAE) calculated from the CEPAS red and blue channels, as well as from AE33 channels with nearest matching wavelengths (BC5 at 660 nm and BC2 at 470 nm) are shown as a time trace in figure C1. The data points are 1-hour averages. The figure C1 also shows the correlation between the AAE measured by the two methods. The concentration from AE33 were converted to absorption by multiplying with the mass absorption coefficient from the instrument manual (10.35 $m^2$/g for BC5 and 14.54 $m^2$/g for BC2). When the BC concentration goes to near zero, the AAE from CEPAS becomes exceedingly noisy, so we have left out the CEPAS data for points where the BC concentration is below 100 ng/$m^3$, as measured by the CEPAS red channel. There are no significant changes observed in the AAE over the measurement period and so no clear correlation is observed either. The average AAE from CEPAS is 0.96, with a standard deviation of 0.46, and the mean from AE33 is 1.46, with a standard deviation of 0.12.*"

**- what is the uncertainty of the instrument for determination of absorption coefficient, BC and NO2 concentration?**

We have added an estimate for the uncertainty of the absorption measurement based on the $NO_2$ calibration into the appendix A.

The addition in appendix A reads (line 350):

"*To estimate the uncertainty of the absorption measured with CEPAS, we consider contribution from the $NO_2$ concentration and absorption cross section used for the calibration and from the power measurement used to scale the CEPAS signals from the different channels. The uncertainty of the power meter is given as 5%. The $NO_2$ concentration is diluted with mass flow controller (FC-785C, Aera) and we estimate the uncertainty of the dilution system to be 5% based on the accuracy of the mass flow controller. Since the $NO_2$ absorption cross section here is based on literature values, we estimate its uncertainty by comparing several spectra (Davidson, et al., 1988; Vandaele et al., 1998; Vandaele et al., 2002; Voigt, et al., 2002) available from MPI-Mainz UV/VIS Spectral Atlas (Keller-Rudek, et al., 2013). The standard deviation of the cross section at the laser wavelengths from the four different spectra gives relative uncertainty of 9.8%, 1.6% and 4.0% for the red, green and blue channels, respectively. The $NO_2$ absorption is further influenced by any wavelength changes of the three lasers. Based on wavelength measurements performed in laboratory conditions on different days, we estimate that the wavelength of the lasers may vary within approximately a range of 1 nm. Allowing each laser wavelength to vary by 0.5 nm to either direction imparts a relative uncertainty of 2.9%, 3.5% and 4.8% for the red, green and blue channel, respectively. The red channel has the largest uncertainty for the absorption cross section, dominated by the differences between different sources. If we use 10% uncertainty for the absorption cross section, combining the other sources addressed above, we get a total uncertainty of 12% for the absorption measured with CEPAS.*"

We have added the references (Davidson, et al., 1988; Vandaele et al., 2002; Voigt, et al., 2002) to the reference list:

*Davidson, J.A., Cantrell, C.A., McDaniel, A.H., Shetter, R.E., Madronich, S., and Calvert, J.G.: Visible-ultraviolet absorption cross sections for NO2 as a function of temperature, J. Geophys. Res. 93, 7105-7112, https://doi.org/10.1029/JD093iD06p07105, 1988.*

*Vandaele, A.C., Hermans, C., Fally, S., Carleer, M., Colin, R., Mérienne, M.-F., and Jenouvrier, A.: High-resolution Fourier transform measurement of the NO2 visible and near-infrared absorption cross-section: Temperature and pressure effects, J. Geophys. Res. 107, ACH 3-1–ACH 3-12, https://doi.org/10.1029/2001JD000971, 2002.*

*Voigt, S., Orphal, J., and Burrows, J.P.: The temperature and pressure dependence of the absorption cross-sections of NO2 in the 250-800 nm region measured by Fourier-transform spectroscopy, J. Photochem. Photobiol. A: Chem., 149, 1–7, https://doi.org/10.1016/S1010-6030(01)00650-5, 2002.*

The uncertainty in terms of the BC concentration is difficult to ascertain, since we did not have a well calibrated method to compare against, nor a convenient way to estimate the true MAC values.

The $NO_2$ concentration estimation as described here is a preliminary description of the method. The accuracy is difficult to comment on with much confidence, since the results, as described here, are dependent on the occasional link to the reference instrument and the accuracy will likely depend on the distance from the nearest anchor point. An independent $NO_2$-free background measurement is planned to be implemented in the future and we believe is required for a fair estimation of the uncertainty.

**3. Technical corrections**
**Line 83: "The light source is an RGB laser module (µRGB module, OptLasers),"**
  **The change of laser diode spectra can affect measurement of NO2 concentration.**
  **Are the laser diodes temperature controlled? How stable are the laser diode emission spectra?**

The baseplate, which each laser is attached to, is kept at a constant temperature with a thermoelectric cooler. We have added a sentence describing this to the first paragraph of section 2 in the revised manuscript (line 88):

*"Each laser diode is connected to a common baseplate, which is cooled with a thermoelectric cooler."*

However the heat flow from each laser to the common baseplate was passive, which potentially leaves a temperature gradient from the lasers to the point where the baseplate temperature is measured at. The laser diodes used here are, in any case, multimode diodes with limited wavelength stability. The wavelength stability was not of high importance when designing the instrument since the BC measurement was the main application target and, as mentioned by the reviewer later on, small wavelength changes are not an issue there. The fact that we could estimate the $NO_2$ concentration was realized only afterwards, which is also the reason why we did not prepare to measure an $NO_2$ free background during the campaign. Based on measurements of the laser spectra in the laboratory on different days, the peak wavelength varied within approximately 1 nm. This is addressed in the new addition to appendix A, which is detailed in the answer to the previous comment.

**Line 115: "uncertainty in laser wavelengths imposes uncertainty in converting the NO2 absorption cross section to absorption"**

In the Appendix A it is stated that during the calibration the laser spectra were measured. In that case there should be no additional uncertainty for the calibration. The aerosol absorption measurements are not affected by small wavelength shifts. More care should be taken when measuring NO2 in the field.

The comment in the main manuscript was indented more as a general comment, since the wavelength can have a substantial effect on the $NO_2$ absorption measurement. However, as discussed above, the laser temperature in this work was not controlled to a high accuracy. The wavelength measurement was not simultaneous with the calibration measurement, because for the calibration measurement, we wanted to keep the instrument at the configuration used for the actual measurements, but the wavelength measurement required small adjustments to the optical configuration to get the laser light to the optical spectrum analyzer. There may well be some wavelength drift going from the wavelength measurement to the actual calibration measurement which could have affected the calibration. The success with the $NO_2$ concentration measurement suggests that the wavelength drift was likely not a major issue here, but we did not explicitly control for it.

**Line 143: "BC concentrations are relatively low"**

Concentrations are low compared to what? You can state average BC concentration.

The average concentration was added to the description of the station.

The sentence in the last paragraph of section 2: "*The station is classified as urban background station and typical BC concentrations are relatively low.*"

was revised to (line 144): "*The station is classified as urban background station and typical BC concentrations are relatively low, with the average BC concentration over the measurement period of approximately 200 ng/m$^3$.*"

**Line 154: "The standard deviations over the whole data set were 0.28, 0.18 and 0.19 Mm-1 for the red, green and blue channels, respectively."**

What was the averaging interval?

The values are the standard deviation for data averaged over each sample exchange, so effectively the averaging period is 26 s. In the original manuscript this was described later on when discussion the Allan deviation, but in the revised manuscript it has been moved to after the sentence quoted above due to other changes in the Allan deviation plot made based on the reviewer 2 comments.

We have clarified this part by revising the sentence in the first paragraph of section 3.1 "*The standard deviations over the whole data set were 0.28, 0.18 and 0.19 Mm$^{-1}$ for the red, green and blue channels, respectively.*"

into (line 160): "*When the data was averaged over each sample exchange cycle of 20 s, the standard deviations over the whole data set were 0.28, 0.18 and 0.19 Mm$^{-1}$ for the red, green and blue channels, respectively. The time step between measurement cycles is 26 s seconds on average, which is slightly longer than the time it takes to complete one cycle, because the background measurements are not included in the data.*"

Related to this point we have also revised the term "*standard deviation*" used at the end of the same paragraph related to figure 2 into "*Allan deviation*" for accuracy (line 175).

**Line 172:** "**Figure also shows BC concentration from two filter-based photometers (AE33 and MAAP) for comparison.**"

      **Which wavelength was used for AE33 BC data?**

The data is taken from the BC6 channel, measured at 880 nm, which the instrument uses to report the BC concentration. We have clarified this in the manuscript by adding the sentence: "*The concentration for AE33 is taken from the BC6 channel, measured at 880 nm.*" into the figure 3 caption.

**Line 229: "Since the NO2 absorption affects the background signals of the measurement …"**

      **Please clarify the term "background signal" as it contains "instrumental" and gas absorption components.**

      **How fast is the drift of the "instrumental background" (Mm-1 per day)?**

We have added a description of this to the beginning of section 3.3. The addition reads (line 238):

"*The background signal, which is measured after the sample flow passes through a particle filter, consists of an instrumental background, which is influenced by factors such as residual absorption by the cell windows, and a gaseous absorption background due to absorption signal from atmospheric gases. The main contributor to the gaseous absorption component at the wavelengths used for the measurements is $NO_2$. The instrumental background can change over time due to effects such as drifting of the laser alignment or accumulating of absorbing particles on the cell windows. The average drift of the instrumental background per day over the measurement period was measured to be approximately 6.5, 10 and 7.7 $Mm^{-1}$, for red, green and blue channels, respectively, and it was observed to be mostly correlated between the channels*."

We have also added a sentence later on to the same paragraph, below equation 1 to tie the original explanation to the terminology adapted from the reviewer comment. The addition reads (line 253):

"*This effectively attempts to account for changes in the instrumental background of the blue channel, according to how the red channel background has changed.*"

**Line 259: "portable instrument"**

      **Please change to "field instrument".**

Corrected as suggested.

**Line 355: "The underlying data is available from the authors upon reasonable request."**

      **Please correct the declaration in accordance with the journal's Data policy.**

The data availability statement has been revised to (line 400):

"*The BC concentration from MAAP and the NO2 reference instrument data are available through the SmartSMEAR (Junninen et al., 2009) web interface available currently at https://smear.avaa.csc.fi/. Other underlying data is available at https://doi.org/10.5281/zenodo.14496882 (Karhu et al., 2024).*"

The references (Junninen et al., 2009) and (Karhu et al., 2024) were added to the reference list:

*Junninen, H., Lauri, A., Keronen, P., Aalto, P., Hiltunen, V., Hari, P., Kulmala, M.: Smart-SMEAR: online data exploration and visualization tool for SMEAR stations, Boreal Environ. Res. 14, 447–457, 2009.*

*Karhu, J., Mikkonen, T., Kuula, J., Virkkula, A., Ikonen, E., Vainio, M., Timonen, H., and Hieta, T.: Dataset for Field-deployable cantilever-enhanced photoacoustic instrument for aerosol light absorption measurement at three wavelengths, Zenodo [data set], https://doi.org/10.5281/zenodo.14496882, 2024.*

**Anonymous Review 2 comments:**

1. **In Section 3.1 the Allan deviation of the absorption coefficient measured by CEPAS is shown. It might be of interest for the reader so see the Allan deviation plot not only for averaged data but with all available data points shown. This would allow the reader to judge the noise of the instrument.**

The Allan deviation has been recalculated from the unaveraged data. The related discussion in the section 3.1 has been revised accordingly. The original text

"*The standard deviation of measurements averaged over each gas exchange cycle is 0.24 Mm-1 or below. The time step between measurement cycles is 26 s seconds on average, which is slightly longer than the time it takes to complete one cycle, because the background measurements are not included in the data.*"

was revised to (line 171): "*The short term noise is 0.58, 0.40 and 0.42 for red, green and blue channels, respectively. The average time step for the raw data used to calculate the Allan deviation is 3.8 s. This includes all the dead time originating from sample exchange and background measurements, as well as signal processing between each measurement step.*"

To reflect the new calculation result, the sentence "*With 1 hour averaging time, the standard deviation is below 0.04 Mm$^{-1}$ for all channels.*" was revised to (line 175) "*With 1 hour averaging time, the Allan deviation is below 0.05 Mm$^{-1}$ for all channels.*"

We also noticed a related misprint in the second paragraph of section 2. The signal block length used to calculate the Fourier transform of the microphone signal should be 1.5 s and not 1 s as was written in the original manuscript.

The sentence "*The channels were measured simultaneously by taking a 1 s block of the microphone signal and calculating its Fourier transform.*"

was revised into (line 109): "*The channels were measured simultaneously by taking a 1.5 s block of the microphone signal and calculating its Fourier transform.*"

2. **In Section 3.2, the correlation between absorption coefficient measurements by CEPAS and AE33 are shown for all CEPAS wavelength. The correlation is very high, but the slopes differ significantly from unity. Since a MAAP instrument is available at the site, the correlation between MAAP and CEPAS should be shown, particularly because the MAAP reports more robust absorption coefficient data than the AE33.**

The correlation plot between MAAP and the red channel of CEPAS has been added to the revised manuscript as described in the response to the third comment of Anonymous Reviewer 1. The correlation against MAAP showed a slight improvement compared to the BC 5 channel of AE33, but the slope still shows clear disagreement, as is apparent also from figure 3. It is, however, difficult to say if this is due to uncertainty of MAAP or CEPAS. A more comprehensive comparison to other standardized methods is, as noted in the general comments of Anonymous Reviewer 2, an important future prospect, but beyond the scope of this article.

3. **Since both CEPAS and AE33 report absorption Angstroem exponents, a comparison of both AAE values would be of high interest. As was shown by Weber et al. (2022), the are large uncertainties associated to the measurement of AAE. The authors mention such uncertainties or differences, respectively, in lines 205 to 207, but without further discussion. This discussion is definitely needed, since the main purpose of multi-wavelength measurements is to derive Angstroem exponents for all aerosol optical parameters.**

An appendix C has been added showing the AAE measured with CEPAS and AE33 and discussing this topic. However, we did not observe clear changes in the AAE over the measurement period, so the conclusion that we can draw from the available data is limited. See the response to the fourth comment of Anonymous Reviewer 1 for details of the revision.

**MINOR ISSUES**
1. **Line 142: Suggested rephrasing: "The instrument was moved to the air quality monitoring station …".**
Corrected as suggested.

2. **Line 159: Please correct "by approximately one order of magnitude …".**
Corrected as suggested.

3. **Line 178: Please correct "assuming an absorption Angstroem exponent of 1.0 ."**
Corrected as suggested.

4. **Line 289: The meaning of the phrase "will slowly flow pass the ... " is not clear, please check.**
The original sentence "*The issue with a direct gas measurement is that the gas will slowly flow pass the micrometer-scale gap around the cantilever, so the gas concentration will slowly equalize between the two sides.*"
was revised into (line 305): "*The issue with a direct gas measurement is that gas can flow pass the micrometer-scale gap around the cantilever, so that the gas concentration will slowly equalize between the two sides of the cantilever.*"

**REFERENCES**
**Weber, P., Petzold, A., Bischof, O. F., Fischer, B., Berg, M., Freedman, A., Onasch, T. B., and Bundke, U.: Relative errors in derived multi-wavelength intensive aerosol optical properties using cavity attenuated phase shift single-scattering albedo monitors, a nephelometer, and tricolour absorption photometer measurements, Atmos. Meas. Tech., 15, 3279-3296, doi: https://doi.org/10.5194/amt-15-3279-2022, 2022.**

Other revisions:

We have additionally revised the reference
*Kuula, J., Karhu, J., Mikkonen, T., Grahn, P., Virkkula, A., Timonen, H., Hieta, T., Vainio, M.: Validation of cantilever-enhanced photoacoustic particle size-resolved light absorption measurement using nigrosin reference particles and Mie-modelling, Submitted to Aerosol Research, 2024.*

into:
*Kuula, J., Karhu, J., Mikkonen, T., Grahn, P., Virkkula, A., Timonen, H., Hieta, T., and Vainio, M.: Validation of cantilever-enhanced photoacoustic particle size-resolved light absorption measurement using nigrosin reference particles and Mie-modelling, Aerosol Research Discuss. [preprint], https://doi.org/10.5194/ar-2024-26, 2024.*